# Characterization of a Bioink Combining Extracellular Matrix-like Hydrogel with Osteosarcoma Cells: Preliminary Results

**DOI:** 10.3390/gels9020129

**Published:** 2023-02-03

**Authors:** Giada Loi, Gaia Stucchi, Franca Scocozza, Laura Cansolino, Francesca Cadamuro, Elena Delgrosso, Federica Riva, Cinzia Ferrari, Laura Russo, Michele Conti

**Affiliations:** 1Department of Civil Engineering and Architecture, University of Pavia, Via Adolfo Ferrata 3, 27100 Pavia, Italy; 2Department of Clinical Surgical Sciences, University of Pavia, Via Adolfo Ferrata 5, 27100 Pavia, Italy; 3Department of Biotechnology and Biosciences, University of Milano-Bicocca, Piazza della Scienza 2, 20126 Milan, Italy; 4Department of Public Health, Experimental and Forensic Medicine, Histology and Embryology Unit, University of Pavia, Via Forlanini 2, 27100 Pavia, Italy; 5Animal Welfare and Radiobiology Service Center, University of Pavia, Via Adolfo Ferrata 5, 27100 Pavia, Italy; 6CÚRAM SFI Research Centre for Medical Devices, National University of Ireland Galway, H92 W2TY Galway, Ireland

**Keywords:** development, characterization, three-dimensional bioprinting, hybrid hydrogel, osteosarcoma, three-dimensional culture

## Abstract

Three-dimensional (3D) bioprinting allows the production of artificial 3D cellular microenvironments thanks to the controlled spatial deposition of bioinks. Proper bioink characterization is required to achieve the essential characteristics of printability and biocompatibility for 3D bioprinting. In this work, a protocol to standardize the experimental characterization of a new bioink is proposed. A functionalized hydrogel based on gelatin and chitosan was used. The protocol was divided into three steps: pre-printing, 3D bioprinting, and post-printing. For the pre-printing step, the hydrogel formulation and its repeatability were evaluated. For the 3D-bioprinting step, the hydrogel-printability performance was assessed through qualitative and quantitative tests. Finally, for the post-printing step, the hydrogel biocompatibility was investigated using UMR-106 cells. The hydrogel was suitable for printing grids with good resolution from 4 h after the cross-linker addition. To guarantee a constant printing pressure, it was necessary to set the extruder to 37 °C. Furthermore, the hydrogel was shown to be a valid biomaterial for the UMR-106 cells’ growth. However, fragmentation of the constructs appeared after 14 days, probably due to the negative osteosarcoma-cell interference. The protocol that we describe here denotes a strong approach to bioink characterization to improve standardization for future biomaterial screening and development.

## 1. Introduction

Three-dimensional in vitro models present a challenging opportunity to advance in tissue engineering (TE) and regenerative medicine [1], as they represent an alternative method that better mimics the real complexity of tissues in vivo, compared to two-dimensional (2D) cultures [2]. The advent of innovative and advanced technologies, such as 3D bioprinting (BioP), allows the production of artificial 3D cellular microenvironments thanks to the controlled spatial deposition of bioinks, i.e., a mix of a biomaterial (usually hydrogel) and biological components (such as cells) [3]. Such bioinks must satisfy the following requirements: printability, appropriate physico-chemical features (i.e., stiffness and viscosity), and biocompatibility [4,5,6].

Therefore, an adequate hydrogel design and characterization are essential to achieve the requirements necessary for 3D BioP and cell culture. In particular, the assessment of printability and shape fidelity is a crucial step in the development of a bioink. As hydrogel filaments are the basic building blocks in BioP, their formation and stacking during the layer-by-layer printing process must be evaluated [7,8,9,10,11]. Once reproducible control over filament deposition is achieved, shape fidelity needs to be assessed in more complex constructs, such as planar or multilayer structures [12,13,14,15,16,17,18]. During or immediately after BioP, hydrogel cross-linking is performed, ensuring the structural integrity of the printed construct. The cross-linking process strongly affects the properties of the bioink, depending on the exposure time and the amount of cross-linker [19,20,21,22,23,24]. Finally, once the 3D construct has been created, cells migrate and proliferate in vitro, generating the biological-tissue model. For this aspect, it is important to guarantee the biocompatibility of the substrate and reproduce the native network with appropriate stiffness and porosity to modulate cell behavior.

Combining printability with cell viability is a challenging task [25]. However, this need can be met by developing customizable inks in which the polysaccharide and protein components of the extracellular matrix (ECM) (i.e., fibrous proteins, glycoproteins, glycosaminoglycans, and proteoglycans) can be suitably cross-linked and mixed to “tune on demand” the mechanical and biological properties of the final construct [26]. The role of the ECM in cell-fate control is well established in the literature: tissue stiffness, such as the ECM biochemical composition, is specific across different organs and between physiological and pathological states [27]. Furthermore, cross-linking strategies are widely used to control the stiffness and structural organization of the final 3D scaffold. However, the most common hydrogels in BioP, such as sodium alginate (SA)-gelatin or methacrylated gelatin, require ionic or UV-based crosslinking strategies involving the introduction of potentially cytotoxic agents/reagents with side effects [28,29]. An important chemoselective reaction is the Diels–Alder cycloaddition, which occurs under mild experimental conditions, suitable for the co-presence of cells, and the kinetics of the cross-linking can be opportunely tuned for different biological applications [30].

Keeping both printability and cell-viability aspects in mind, a new protein–polysaccharide hybrid biomaterial, which is promising as a bioink, was developed. It is based on gelatin (GE) and chitosan (CH) functionalized with methylfuran groups and cross-linked by Diels–Alder cycloaddition with a commercial Star-PEG functionalized with maleimide groups as a dienophile (Star-PEG-MA) [31,32]. Compared to traditional biomaterial inks, i.e., hydrogel without biological components [33], the advantages of the functionalized biomaterial employed in this study include the control of biological and mechanical features by tunable and cell-safe crosslinking. Furthermore, it is well established in the literature that the microenvironment recreated by CH is suitable for cell growth [34] and that GE is a natural polymer that has specific amino-acid residues, such as Arginyl-glycyl-aspartic acid (RGD) [35], which is an ECM adhesion sequence [36]. For this reason, CH–GE hybrids are considered promising materials for TE applications [37,38].

This GE–CH biomaterial ink was previously tested through rheological tests, chemical characterization, and cellular tests with U87-MG and Gli36ΔEGFR-2 glioblastoma cells [31,39]. However, to overcome the current state-of-the-art of this biomaterial, its characterization during the BioP steps, through the application of an engineering and standardized approach, was necessary. Furthermore, a detailed study of cross-linking time-based printability, including differential fiber dimension, inter-operator feasibility, and cellular versatility, was lacking.

In order to investigate these biomaterial aspects, in this work, we propose a protocol divided into three interrelated steps: pre-printing, 3D bioprinting, and post-printing. For the pre-printing step, the hydrogel formulation and its reproducibility were evaluated. Specifically, we analyzed the effect of the amount of cross-linker and the biomaterial behavior over time after the start of the cross-linking process. Next, to evaluate the protocol’s reproducibility, we compared hydrogel batches prepared by different operators and for each condition considered, several repetitions were printed. For the 3D-bioprinting step, the printability performance of the hydrogel was assessed through qualitative and quantitative tests, according to the literature, using an extrusion-based 3D bioprinter. Finally, for the post-printing step, the hydrogel biocompatibility was investigated, in a preliminary way, by developing a 3D osteosarcoma model using UMR-106 cells.

## 2. Results and Discussion

### 2.1. Layer Stacking or Merging: Temperature and Cross-Linking Time Effect

The material was suitable for printing grids with a good resolution from 4 h after the addition of the Star-PEG-MA (Figure 1b). The pressure increased during the printing period from about 10 kPa, for the first prints, to about 65 kPa, starting from 5 h post-cross-linking. This behavior was due to an effect of polymerization and, in particular, to the temperature’s influence on the biomaterial ink, which, composed of gelatin, is conditioned by its thermos-sensitivity [37]. Thus, the material had good printability in the first prints, i.e., immediately after it was removed from the incubator, while over time, the material stiffened and the filaments were more deformed. Therefore, as a subsequent test, printing with the extruder heated to 37 °C was analyzed to understand whether the properties of the material changed over time due to polymerization or due to hardening caused by the decrease in temperature. In this second condition, the material was suitable for printing from 4 to 24 h after adding the Star-PEG-MA cross-linking agent. The printing pressure was about 50–55 kPa and remained constant over time (Figure 1bii).

### 2.2. Drop or Fiber Formation: Cross-Linking Optimization

In the drop/fiber formation test, the morphology of the hydrogel exiting the printer nozzle was evaluated in relation to the ability to form fibers or droplets.

As shown in Table 1 and Figure 2, hydrogel samples A and B passed the test with fiber formation in the pressure range of 55–80 kPa, with similar features. Below this interval, the material formed agglomerates at the nozzle exit and, above, it formed increasingly smooth and less deformed fiber with excessively high pressures incompatible with any cellular component. The results of these tests strongly depend on the hydrogel’s rheological properties [31] and composition [8,9,11]. Viscosity has been suggested as a predictor of potential filament formation rather than droplets and self-supporting structures. In the specific case of gelatin-based hydrogels, as the concentration increases, three states of gelling can be observed for the printed hydrogel: under-gelation, proper-gelation, over-gelation [9]. When the hydrogel was printed with an under-gelation status, it would show droplet morphology at the nozzle tip, resulting in a poor shape fidelity after printing; under proper gelation conditions, uniform filaments and self-supporting structures were created; however, when the hydrogel was in a state of over-gelation, it would easily show a fractured morphology, resulting in irregular filaments and structures. Therefore, in agreement with the literature, our hydrogel showed signs of under gelation, with the formation of drops less than 4 h after the addition of the Star-PEG-MA. Once the polymerization had taken place, it showed over-gelation behavior with the formation, initially, of an accumulation of hydrogel at the exit of the nozzle and, subsequently, the formation of irregular fibers that seemed to become gradually smoother due to the high printing pressure. Sample C, however, required significantly high pressures for extrusion (90–105 kPa). The cause was identified in the increased concentration of cross-linking agent, which led to a higher degree of polymerization translated into increased hydrogel viscosity. However, such high pressures are incompatible with 3D BioP, due to the potential damage to the cellular component during the printing process.

Consequently, the hydrogel formulation of sample C was abandoned, and quantitative tests were applied only to Sample A and B.

### 2.3. Quantitative Printing Performance: Operator and Protocol Repeatability

The filament dimensions obtained were greater than the ideal size, defined by the nozzle diameter (0.41 mm). The filaments’ distance was consequently less than the target value. Finally, the P_r_ demonstrated values slightly greater than 1, suggesting an irregular geometry.

The results of the repeatability analysis for the various parameters are shown in Figure 3. The data obtained from the tests on Sample B do not differ excessively from those obtained from the tests performed on hydrogel Sample A. In fact, no statistically significant difference was found. Therefore, the protocol performed from different operators was confirmed as repeatable.

These results are considered promising as the resulting printing performance was better than that obtained in previous characterizations with common materials in BioP, such as SA-GE hydrogels [40]. With SA-GE hydrogels, a 1-mm filament size and a 1.5-mm minimum filament distance were obtained, compared with the size of 0.7 mm and the resolution of 1 mm obtained in this work.

### 2.4. 3D Cultures

#### 2.4.1. Microscopical Observation of Cell Growth in 3D Models

Microscopic observations of the constructs at the established time points (Figure 4) evidenced, at the first observation time (t = 0), a homogeneous cell density with maintenance of the 3D structure for both the evaluated cell concentrations (7 × 10^5^ cells/mL, 6 × 10^6^ cell/mL).

At the lowest cell concentration (7 × 10^5^ cells/mL), a progressive emptying of the construct and an absence of cell-clone formation was observed up to 14 days (data not shown). For this reason, the bioink concentration of 7 × 10^5^ cells/mL was no longer taken into consideration.

The constructs printed with 6 × 10^6^ cell/mL observed a week after their printing showed a reduced cell density, probably due to the stress related to the printing process. At this time point, it was also possible to observe the presence of a few clones with dividing cells. The size and number of clones increased at the subsequent time-point observations, indicating the proliferation of the cells embedded in the hydrogel, as also confirmed by the imaging results.

However, a large fragmentation of the 3D structures and, in some samples, the halving of the construct, was observed. The observation at 21 days was unreliable for the 3D cell-culture study and did not allow a correct evaluation of cell growth. It was not possible to keep the constructs in the culture for a period longer than 28 days due to a complete loss of structure.

#### 2.4.2. Cell Impact on Shape Fidelity and 3D Structures

The presence and growth of cells in a bioink is a highly relevant aspect for the BioP procedure, but often, in studies on 3D bioprinted models, the impact that cells have on the chemical-physical properties of the hydrogel is neglected. Thus, the effect of cellular interference on the biomaterial behavior reported in this work is a further innovative aspect. This impact depends on several factors, such as: the volume occupied by the cells and their metabolic state, the type of cells and the encapsulation density [41], the chemical interaction between cells and the hydrogel employed, the ECM remodeling, and the induction of specific cell responses (i.e., differential cell receptor and ECM-factor expression or secretion). Therefore, the cells in the bioink occupy a specific volume precluded by the hydrogel components with possible repercussions for: (1) the hydrogel swelling and on (2) the viscoelastic properties. To reduce these two effects, in this work, the cells were added after the polymerization was completed and different cellular concentrations were examined. Furthermore, cellular influence was investigated by comparing bioprinted constructs with biomaterial ink or bioink.

The constructs composed of biomate”Ial ’nk and bioink are shown in Figure 5. The first type of sample was intact after 3 days and there was an initial fragmentation of the filaments only at 7 days after printing; the constructs maintained shape fidelity up to 21 days, with disintegration at 25 days. For the sample composed of bioink, the internal grid filaments were broken in some constructs after 3 days of printing, with an increase in the number of fragmented constructs at 7 days. This disintegration of the 3D structure progressed over time: at 21 days, most of the constructs composed of bioink were fragmented, with a loss of geometry, or had a completely decomposed structure. In general, the constructs printed with biomaterial ink were more resistant in culture, with a higher number of intact geometry constructs, at the same time point, compared to the structures composed of bioinks. In particular, from the experiments carried out, it seemed that the cell type chosen, i.e., UMR-106, negatively interfered with the maintenance of the 3D structure of the construct, allowing it to maintain shape fidelity for up to 14 days.

It has been hypothesized that the cells of the UMR-106 line can destroy the matrix in which they are encapsulated, without implementing a remodeling process to produce new ECM self [42,43]. This effect could also be related to the characteristic infiltrative growth of osteosarcoma [44]. Further studies will investigate this hypothesis.

Furthermore, the changes in culture medium required every 2–3 days for the cell line were used to increase the probability of fragmentation of the internal filaments of the structure, which are subject to greater stresses.

The impact of the UMR-106 cells on the maintenance of the 3D structure over time was further investigated using a concentration of 2 × 10^6^ cells/mL, which was an intermediate value between the previous those of the concentrations evaluated (7 × 10^5^ and 6 × 10^6^ cells/mL).

The growth of the UMR-106 cells (Figure 6) was not comparable with the results previously obtained at different cell concentrations; larger clones were observed at 14 days, as well as a greater consistency in shape fidelity for three-dimensional structure over time. Therefore, for the hybrid hydrogel, studying a lower cell concentration leads to less cell interference, with a consequent reduction in the probability of fragmentation of the internal filaments and disintegration of the 3D structure.

### 2.5. Limitations

Despite the success of the proposed characterization protocol and the promising results obtained, some limitations emerged. In particular, one of the main limitations concerns the biomaterial stability with the selected geometry and fiber dimensions during the culture period with UMR-106 cells. This aspect is crucial for TE and regenerative medicine [41] and it is a specific bioink-based behavior. For each bioink, this aspect is influenced by the cellular identity, the culture conditions, the ad hoc CAD design, the fiber dimensions, and the construct in general. To increase its versatility, it is necessary to investigate the cause of this dissolution behavior in the case of UMR-106 cell cultures. The hypothetical causes include a high-interaction hydrogel-culture medium which causes its dissolution and, more likely, interference due to the cell line, UMR-106. This second hypothesis is the most accredited because similar problems did not occur in the previous studies with U87-MG and Gli36ΔEGFR-2 [31,39]. Further studies will focus on whether UMR-106 cells could destroy the matrix in which they are encapsulated, without implementing a remodeling process to produce a new ECM self. Furthermore, to overcome this limitation for applications with UMR-106 cells and long periods of culture (over two weeks), the biomaterial formulation will be optimized to make it more rigid and resistant through the increase in cross-linker degree and the modulation of the polysaccharide = −protein ratio.

Another important limitation is that as a gelatin-based bioink, the hydrogel is affected by thermos-sensitivity, as mentioned in Section 2.1. Consequently, the printing temperature has a significant impact on the hydrogel’s printability and construct conditions after the printing process. Furthermore, the ideal printing conditions would also require a temperature-controlled printing area. In fact, only temperature variation in the laboratory environment causes changes in the printing pressures required to extrude the material.

Finally, these results were obtained through the exclusive use of the INKREDIBLE+ printer. To confirm them for extrusion-based technology in general, it would be useful to repeat these characterization tests with other bioprinter models.

## 3. Conclusions

The focus of this experimental work was to characterize a new biomaterial to be applied in the BioP field, in order to define a standardized protocol. A hydrogel based on gelatin and chitosan functionalized with 5-methylfurfural and cross-linked with Star-PEG-MA was used. The following methodological variables were considered: the quantity of the cross-linker, printability-performance assessment as a function of the temperature and cross-linking time, the biocompatibility tested by UMR-106 cells and the maintenance in culture of the 3D constructs.

This hybrid hydrogel showed good extrusion and printing resolution and a dependence on the cross-linking time and temperature. Furthermore, it was shown to be a valid printable biomaterial for the growth of UMR-106 cells in a 3D-bioprinted construct. Regarding the maintenance of the structure, fragmentation of the constructs appeared after 14 days, probably due to the negative interference of the cellular component.

Further studies will be performed to achieve longer incubation times and to investigate the kinetics of cell proliferation and the degradation of the gelatin and chitosan matrix as possible causes of UMR-106-cell-line interference.

## 4. Materials and Methods

### 4.1. Hydrogel Formulation

The hybrid hydrogel formulation was composed of methyl-furan functionalized gelatin (GE–MF) and methyl-furan functionalized chitosan (CH–MF). These polymers (Sigma-Aldrich, Milan, Italy and Carbosynth Ltd., Compton, UK) were functionalized as already reported in previous studies [31] and then freeze-dried and stored at −20 °C before their usage. The GE–MF (165 mg) and CH–MF (85 mg) were dissolved in 4.75 mL of PBS at 7.4 pH. The solution was immersed in a laboratory bath at 37 °C for about 1 h and vortexed until complete dissolution. The 4-arm-PEG10K-maleimide (Star-PEG-MA) (17.5 mg) was dissolved in 0.35 mL of PBS at 7.4 pH at room temperature (RT). A total of 250 μL of the Star-PEG-MA solution was taken and added to the gelatin–chitosan (GE–CH) polymers. The hybrid solution GE–CH (Figure 7) was left for 3 h at 37 °C to allow hydrogel-network formation.

### 4.2. Three-Dimensional Bioprinter

To print the hybrid hydrogel, the extrusion-based 3D bioprinter Cellink INKREDIBLE +  (Cellink AB, Göteborg, Sweden), equipped with two printheads (PHs), was adopted. Printheads can be heated up to a maximum of 130 °C. To ensure the sterility of the printing chamber, the INKREDIBLE+ was equipped with patented Clean Chamber Technology, a UV LED curing system (365 and 405 nm), and a high-efficiency particulate air filter, HEPA H13. The process started with the design of structures as CAD files, after which the virtual 3D geometry model was translated, through a slicing software, into machine instructions, i.e., the G-code; eventually, constructs were 3D-printed. The hybrid hydrogel was transferred and printed using a plastic cartridge and a conical 0.41-mm nozzle.

### 4.3. Bioprinter Set-Up

Before starting the printing process, the bioprinter was calibrated. First, the XYZ axes were homed to position PHs in the center of the printing bed. Next, Z axis was calibrated to record the distance between the nozzle and the printing bed, and, finally, the printing temperature and pressure, which ensured the proper material flow, were set (Table 2).

### 4.4. Scaffold Design and 3D Printing

Using Autodesk Inventor^®^ software (2021), we created a 12-millimeter-diameter cylindrical structure (Figure 8a) and a square structure with sides measuring 10 mm (Figure 8b). Height of both structures was set at 0.70 mm. Next, the CAD model was sliced using Slic3r, an open-source slicing software. During the slicing process, some parameters were defined, such as layer height, number of perimeters, printing speed, and infill percentage. The infill parameter is crucial for 3D BioP as it determines the distance between two adjacent fibers, i.e., it defines the pore size of the scaffold. The pore size is a crucial factor, as it guarantees the correct supply of nutrients and oxygen to cells, guaranteeing their growth. After the slicing, we obtained for the cylindrical structure a geometry with a 30% infill, while for the square structure, we obtained a geometry with a 20% infill. Both structures consisted of two layers 0.35 mm high. The printing speed was fixed at 450 mm min^−1^, which, combined with pauses at each change in extruder trajectory, allowed the correct adhesion of the hydrogel on the Petri dish. The G-code was created and constructs were 3D-printed. We printed the square scaffold for the printability-resolution tests, while the cylindrical structure was used for the biocompatibility tests. For this second case, we printed scaffolds using both the bioink and the biomaterial ink.

### 4.5. Assessment of Printing Performance

To assess the printing performance, we evaluated the extrudability and the printability of the biomaterial ink using an extrusion-based 3D bioprinter, following qualitative and quantitative protocols implemented by Paxton et al. [10] and Schwab et al. [41], respectively (Figure 9).

#### 4.5.1. Assessment of Printing Performance: Qualitative Protocol

The first protocol consists in performing two qualitative tests described below.

Layer stacking or merging: this test of qualitative printability assessment is designed to evaluate the hydrogel’s ability to form self-supporting 3D structures. Fibers must be able to stack layer-by-layer without merging. A simple test for layer stacking consists of generating, through BioP, multilayer grids (typically planar structures composed of 1–2 layers) [41]. The realization of grids with well-defined pores indicates a passed test, while a collapsed structure indicates a failed test [8,9,11] (Figure 9ai).

Drop or fiber formation: this simple screening method makes it possible to analyze the material’s ability to form fibers, rather than droplets. The proposed test for the filament morphology consists of a visual screening, i.e., in the observation of the ink morphology exiting the nozzle by slowly increasing the air pressure until the hydrogel begins to be extruded steadily [7]. The formation of a liquid droplet indicates a failed test, while the formation of a thin filament indicates a passed test (Figure 9aii).

Only materials that pass both tests should be considered for further development.

##### Layer Stacking or Merging: Cross-Linking-Time and Printing-Temperature Effect

To investigate the cross-linking time, structures were printed at intervals of 20 min, from 3 h up to 5 h and 24 h after the addition of the cross-linker agent, Star-PEG-MA. In order to also evaluate the printing temperature effect, the material was printed on a slide cover both at RT and by heating the printer extruder to 37 °C. Temperature was kept constant throughout the printing process. We 3D-printed five scaffolds for each time point after the cross-linker addition.

Only the conditions that ensured the realization of grids with well-defined pores were considered for the next steps.

##### Drop or Fiber Formation: Cross-Linking Optimization

The composition of the hydrogel was optimized to obtain constructs with greater structural stability and that lasted longer over time. In this regard, printability tests were carried out with three samples of 5 mL each (A, B, C) of 5% GE-CH-based hydrogel cross-linked with Star-PEG-MA, prepared according to the method described in Section 4.1. The concentrations used for each sample are indicated in Table 3. For the evaluation of the protocol’s repeatability, Sample B was prepared by an operator different from the one who prepared the other two samples. In sample C, the GE−MF and CH–MF concentrations remained unchanged, while the concentration of the cross-linking agent, Star-PEG-MA was increased by 1.5 times compared to the protocol previously developed.

The analysis was carried out for all the samples 5 h after the addition of the cross-linking agent, increasing the pressure by 5 kPa every 5 s, starting from 0 kPa up to 125 kPa [7]. The printing pressure was varied to identify the optimal pressure for the bioprinting process. Only materials that allowed the formation of a fiber instead of a droplet, at pressures suitable for cell printing, were considered for the following steps.

#### 4.5.2. Assessment of Printing Performance: Quantitative Protocol

To assess the printing performance, we printed square scaffolds (Figure 8b) and evaluated the printability, including shape fidelity and filament characterization, following the protocol proposed by Schwab et al. [41] (Figure 9bi). The protocol consists in performing three quantitative tests, described below.

Filament size: the homogeneity of the filaments was evaluated by measuring the filament diameter ($d_{1}$, $d_{2}$, $d_{3}$, …, $d_{i}$; Figure 9bi,ii). Equal diameters denote a homogeneous filament. The filaments’ average size ($\bar{d}$) was calculated as the ratio between the sum of the diameters taken from the image analysis in different positions of the construct ($d_{i}$) and the total number of measurements (N).
(1)$$\bar{d}=\frac{\sum_{i=1}^{N}d_{i}}{N}\;\;\;\;$$Filament distance: the surface tension between the biomaterial and the printing support (in this case, the Petri dish), as well as between each layer of the material, can cause the merging of adjacent filaments (Figure 9bi,ii). Therefore, an important parameter to define the resolution of our material is the minimum distance that it is able to guarantee between the grid filaments. The mean distance ($\bar{D})$ between the filaments was calculated as the ratio between the sum of the distances taken from the analysis of images in different positions of the construct ($D_{i}$) and the total number of measurements (N).
(2)$$\bar{D}=\frac{\sum_{i=1}^{N}D_{i}}{N}$$Pore geometry or printability index: the optimal shape of pore geometry is rectangular, which indicates the ideal filament (Figure 9bi,ii). This evaluation was performed using the printability index ($P_{r}$) [41], calculated as:(3)$$P_{r}=\frac{L^{2}}{16A}\;$$
where *L* and *A* are pore perimeter and area, respectively. The *Pr* index is higher than 1 for irregular structures, 1 in case of perfect square pores, and lower than 1 for circular pores.

##### Quantitative Printing Performance: Protocol and Operator Repeatability

For the repeatability analysis, printing tests of the samples which passed the qualitative tests were carried out. For each sample, three structures were printed at 30-min intervals from 3 h up to 5 h after the addition of the Star-PEG-MA.

Three-dimensionally printed constructs’ images were acquired, and ImageJ software (National Institutes of Health, United States) was used to extract the parameters suggested by protocol, i.e., filament size, filament distance, and printability index. All results are expressed as mean ± standard deviation (SD).

The SDs were calculated and ANOVA tests were applied to evaluate the biomaterial printing repeatability, within-batch and operator-to-operator, respectively.

### 4.6. Cell Culture

Rat-osteosarcoma cell line (UMR-106, European Collection of Authenticated Cell Cultures, Salisbury, Wiltshire, UK) was grown under adhesion conditions in T75 culture flasks with Dulbecco’s Modified Eagle’s Medium, Lonza (DMEM), supplemented with 10% fetal bovine serum (FBS) (Euroclone), and 1% gentamicin (Euroclone), at 37 °C in a humidified 5% CO_2_ atmosphere.

### 4.7. Three-Dimensional Bioprinting Procedure and Biocompatibility

The GE–CH solution was prepared following the protocol reported in Section 4.1. Before starting the printing process, the bioprinter was placed under a sterile hood and the UV light was turned on for 1 h to sterilize all the materials and surfaces. The cross-linked hybrid hydrogel was left for 30 min under UV light for further sterilization. Three cell concentrations were tested: 7 × 10^5^, 2 × 10^6^, 6 × 10^6^ cells/mL.

At about 80% of confluence, cells were detached by trypsin-EDTA treatment and, thereafter, the cells were syringed to avoid the presence of clumps and counted in Burker’s chamber. Predetermined aliquots to be inserted into the hydrogel were centrifuged at 1200 rpm for 10 min. The pellet was then resuspended in 170 µL of DMEM for the higher cell concentration and 200 µL of DMEM for the intermediate and lower concentrations. Cells were sucked into a 1-milliliter syringe and mixed with 800 µL of hybrid hydrogel by connecting the two syringes with a luer connector (Figure 10). The bioink was transferred into a 5-mL bioprinter cartridge and then subjected to a centrifuge pulse at a maximum of 1000 rpm to eliminate every bubble formed during the mixing procedure. Each sample was bioprinted following the geometry previously described in Section 4.4.

To evaluate the impact of cells on the chemical–physical properties of the hydrogel, such as the constructs’ shape fidelity, the following conditions were compared: constructs formed by hydrogel (800 µL) with DMEM (200 µL) and constructs formed by hydrogel (800 µL) with UMR-106 cells (6 × 10^6^/200 µL).

After printing, 3 mL of DMEM medium was added to the constructs (Figure 10) and the Petri dishes were maintained at 37 °C in a humidified 5% CO_2_ atmosphere. The culture media were refreshed every 2–3 days.

### 4.8. Microscopy Analysis

Cell proliferation and distribution within the 3D construct were observed and monitored by a fluorescence microscope (Olympus CX41 with UV light at 350 nm) at different time points (0, 24 h, 7, 14, and 21 days after printing). Three constructs were fixed and observed for each observation time.

For microscopic observation, cells were stained with Hoechst 33342 (Thermofisher, Monza, Italy), a DNA-intercalating dye that is excited by UV light at about 350 nm and has an emission spectrum around 461 nm, producing a blue/cyan fluorescence.

During the first microscopic observation (Figure 4), we noticed slight autofluorescence of the hydrogel and the tendency of the polymeric structure to retain the dye molecule used in the hydrogel mesh. To minimize these effects, we optimized the staining protocol for the observation of constructs at intermediate concentrations (2 × 10^6^), shown in Figure 6. The resulting data in the biocompatibility experiments still represent an intra-experiment-consistent dataset.

The staining protocol required the following steps: (1) removal of the DMEM medium; (2) construct fixation with 70% ethanol, for 20 min; (3) ethanol removal followed by 30 min incubation at 37 °C in Hoechst 33342 dye (10 µg/mL); (4) dye removal and construct washing 3 times in DMEM for 5 min each; (5) transfer of the construct to the specimen slide for the microscopic observation.

## Figures and Tables

**Figure 1 gels-09-00129-f001:**
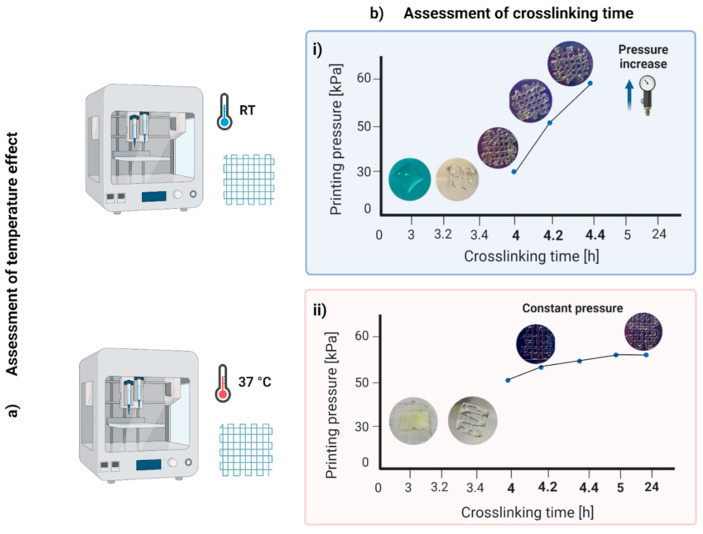
Layer-stacking or -merging results: assessment of (**a**) temperature and (**b**) cross-linking time effect. (**i**) The hybrid hydrogel printed at room temperature (RT) was suitable for printing from 4 to 4.4 h after the addition of the Star-PEG-MA; (**ii**) hybrid hydrogel printed at 37 °C was suitable for printing from 4 to 24 h after the addition of the Star-PEG-MA.

**Figure 2 gels-09-00129-f002:**
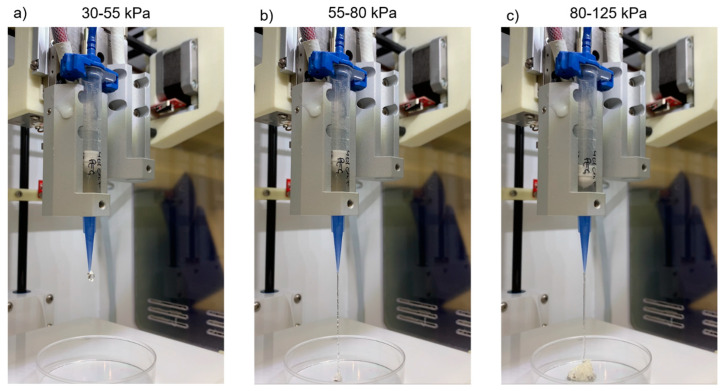
Results of the droplet/fiber formation test for Sample A. (**a**) In the pressure range of 30–55 kPa, hydrogel accumulation occurred at the nozzle outlet; (**b**) at 55–80 kPa, the formation of irregular fibers was observed; (**c**) at 80–125 kPa, regular/smooth fibers were formed.

**Figure 3 gels-09-00129-f003:**
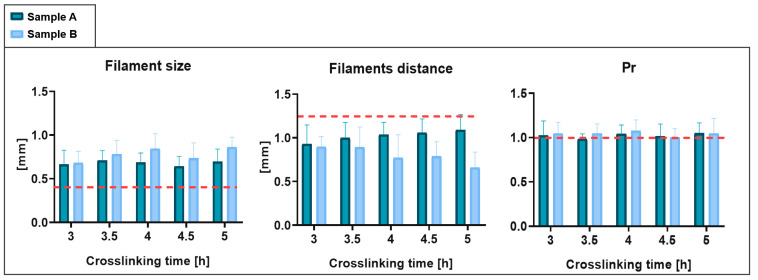
Quantitative printing-performance results: operator and protocol repeatability. The results obtained with Sample A and with Sample B are compared. The dashed red line indicates target values.

**Figure 4 gels-09-00129-f004:**
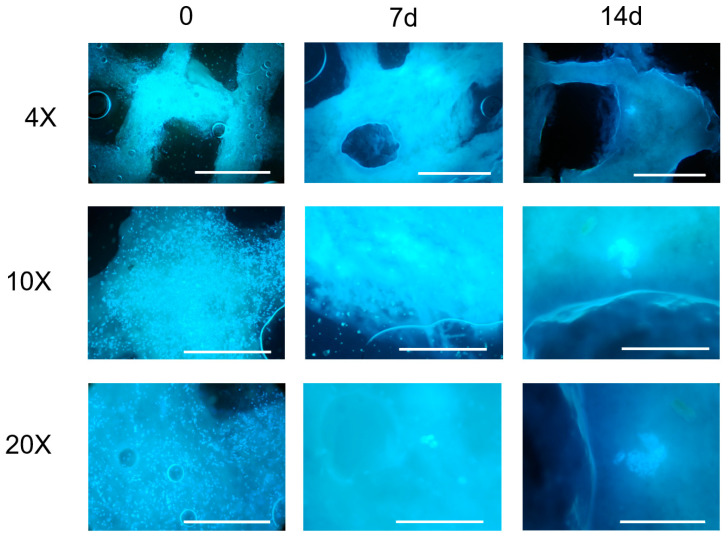
Images from fluorescence microscope (Olympus CX41) at different magnifications (4×, 10×, 20×) of the constructs printed with an initial concentration of 6 *×* 10^6^ UMR-106 cells/mL at different time points. Nuclei stained by Hoechst 33342. Scale bar 1 mm (4×), 500 μm (10×), 250 μm (20×).

**Figure 5 gels-09-00129-f005:**
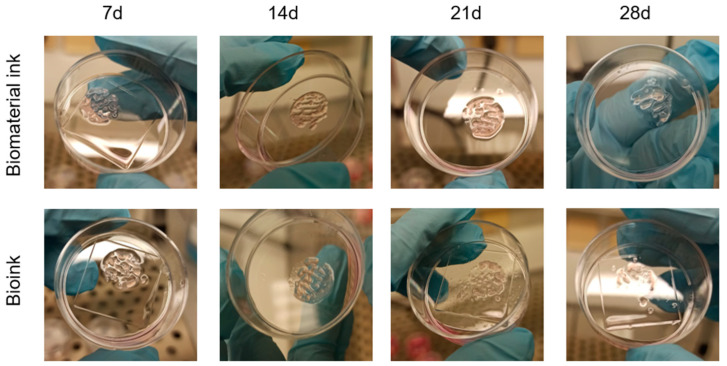
Comparison between the constructs printed with biomaterial ink and bioink at different time points after printing.

**Figure 6 gels-09-00129-f006:**
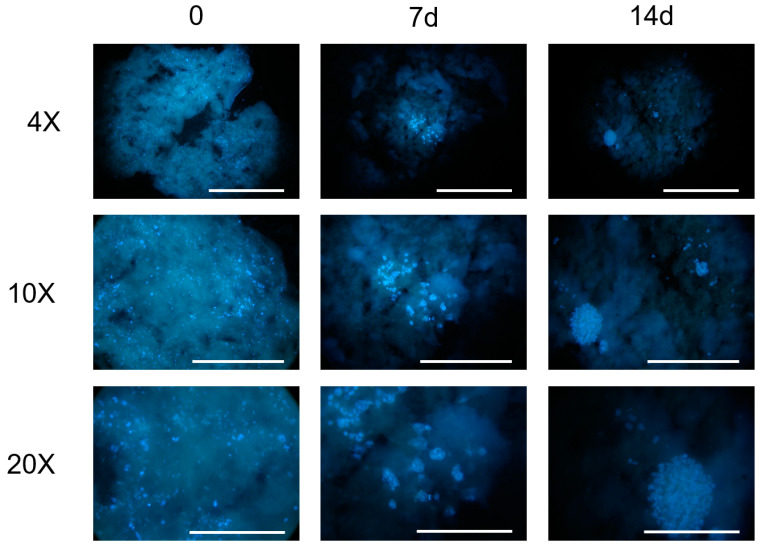
Images from fluorescence microscope (Olympus CX41) at different magnifications (4×, 10×, 20×) of the constructs printed with an initial concentration of 2 *×* 10^6^ UMR-106 cells/mL at different time points. Nuclei stained by Hoechst 33342. Scale bar 1 mm (4×), 500 μm (10×), 250 μm (20×).

**Figure 7 gels-09-00129-f007:**
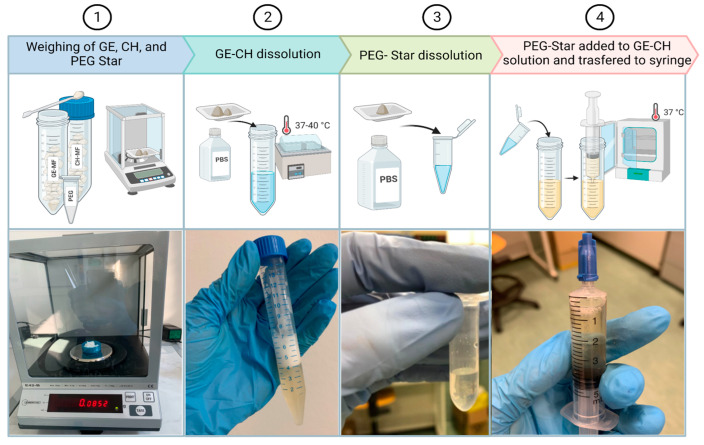
Hybrid-hydrogel preparation. The process started with (**1**) the freeze-dried materials, which were weighted according to the protocol. Subsequently, (**2**) gelatin (GE) and chitosan (CH) were dissolved in PBS, (**3**) the weighted Star-PEG-MA was also dissolved in its PBS aliquot, and, finally, (**4**) the two solutions were combined, transferred into a syringe, and placed in an incubator at 37 °C until completely cross-linked.

**Figure 8 gels-09-00129-f008:**
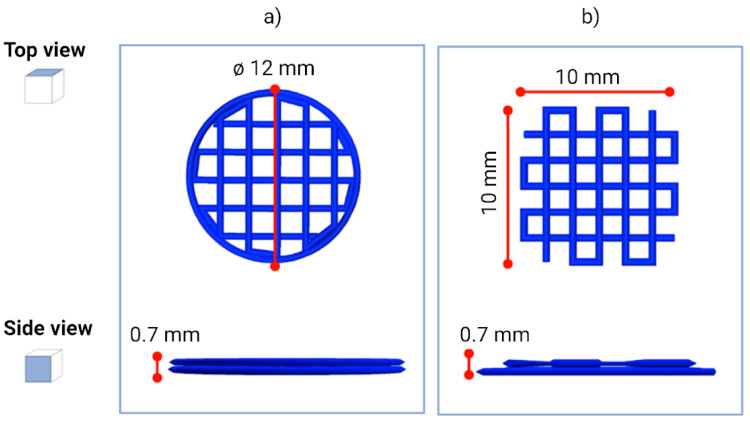
Design of the different 3D geometries used in the study. Schematic representation of the (**a**) cylindrical- and (**b**) square-structure CAD design.

**Figure 9 gels-09-00129-f009:**
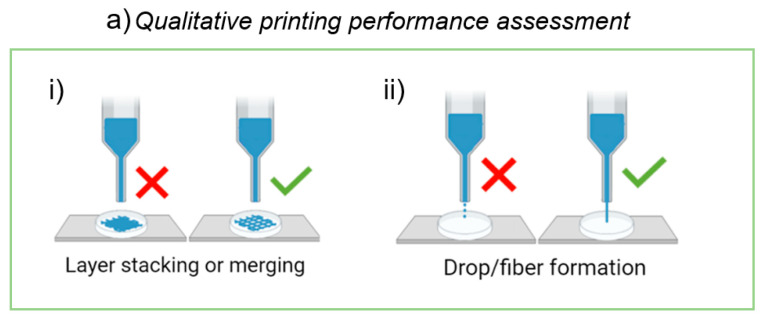
Assessment of printing performance. (**a**) Schematic representation of the proposed method to assess biomaterial ink’s qualitative printability. Initial screening of ink formulations to establish the ability to (**i**) produce grids with well-defined pores and (**ii**) fiber formation, as opposed to droplet formation. (**b**) Schematic representation of the proposed method to assess biomaterial ink’s quantitative printability. (**i**) Protocol for the assessment of printability and shape fidelity proposed by Schwab et al. [41]. (**ii**) Application of the protocol to our 3D virtual model. Scale bar 1 mm.

**Figure 10 gels-09-00129-f010:**
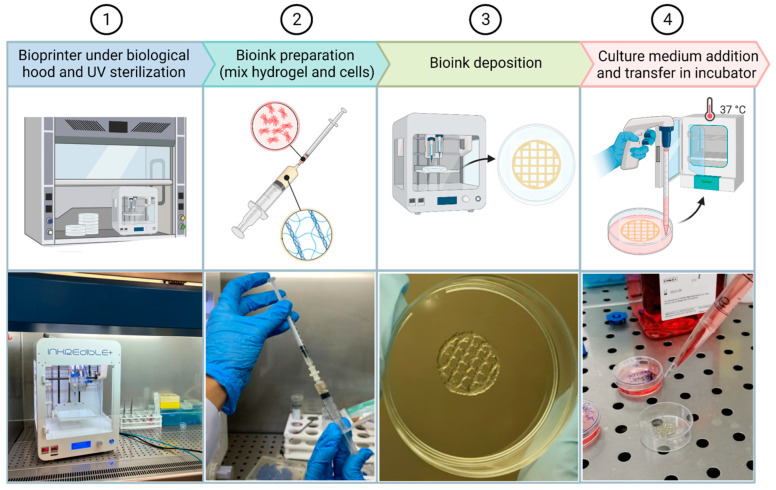
Three-dimensional-bioprinting procedure. It starts with (**1**) the positioning of the bioprinter under a biological hood and UV light sterilization. (**2**) Next, bioink is prepared by mixing cells and hydrogel and (**3**) finally printed. (**4**) Culture medium is added and 3D structures are placed inside the incubator.

**Table 1 gels-09-00129-t001:** Drop or fiber formation test results.

	Pressure (kPa)	Results
Sample A	30–55	Hydrogel accumulation at the nozzle outlet
	55–80	Irregular fiber formation
	80–125	Regular/smooth fiber formation
Sample B	30–55	Hydrogel accumulation at the nozzle outlet
	55–80	Irregular fiber formation
	80–125	Regular/smooth fiber formation
Sample C	30–90	Hydrogel accumulation at the nozzle outlet
	90–105	Irregular fiber formation and break
	105–125	Continuous irregular fiber formation

**Table 2 gels-09-00129-t002:** Summary of printing parameters used for the study.

Three-Dimensional-Bioprinter Operational Variables	
Extrusion pressure (kPa)	50–60
Conical nozzle diameter (mm)	0.41
Printing speed (mm/min)	450
Printing temperature (°C)	RT—37

**Table 3 gels-09-00129-t003:** Composition of the hydrogel samples prepared for qualitative and quantitative printing performance tests.

Sample	GE-MF (mg)	CH-MF (mg)	Star-PEG-MA (mg)
A	165	85	17.5
B	165	85	17.5
C	165	85	26.3

## Data Availability

Data will be made available on request.

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
