# Peer review of "Characterization of a Bioink Combining Extracellular Matrix-like Hydrogel with Osteosarcoma Cells: Preliminary Results"

_gels, 2023, doi:10.3390/gels9020129_

Round 1

Reviewer 1 Report

This manuscript presents some preliminary results on the characterization of a bioink able to produce an extracellular matrix- like hydrogel with appropriate properties for osteosarcoma cells combination. The technical and experimental details are clearly discussed and illustrated with convenient figures. All sections are correctly presented, and the most important details are rightly discussed in a clear format.

I have no scientific comments, but one possible concern is related to the commercial aspect of the manuscript. Everything is referred to the use of 3D bioprinter Cellink INKREDIBLE+®. Authors declare no conflict of interest, but legal requirements would be satisfied concerning the exclusive use of that bioprinter. I have no expertise in the legal aspect, but I have some complain experience as reviewer. I also declare that I have no conflict of interest. I would su, but at lest some comment about the use of this bioprinter in comparison to similar equipment would be added to the limitation section. Figure 8 would be eliminated to ameliorate the commercial trend detectable in the manuscript.

Reviewer 2 Report

This research is interesting. However, there are many papers on ECM-based biomaterial for 4D tumor models. The authors should introduce the concept in detail and explain each material. Without the section, the strength of this study is not clear. Taken together, major revision should be made. This manuscript would be de-considered only when all the comments were responded.

1. Introduction or Discussion

What is the novelty of this study? The authors should introduce the concept and discuss the novelty by comparing research papers using representative biomaterial to claim it. The reviewer suggests the references be added.

Review (for concept)

Cancers 202012(10), 2754

Tissue Engineering Part B: Reviews.Jun 2010.351-359.http://doi.org/10.1089/ten.teb.2009.0676

Research papers

Chitosan doi.org/10.1016/j.biomaterials.2010.03.062

Gelatin Tissue Eng. Part C Methods 201925, 711–720. https://doi.org/10.1089/ten.tec.2019.0189

Collagen doi.org/10.1016/j.biomaterials.2020.119853

Alginate Tissue Engineering Part C: Methods.Jul 2016.708-715.http://doi.org/10.1089/ten.tec.2015.0452

HA doi.org/10.1016/j.ijbiomac.2020.10.095

2. Biological functions should be investigated.

Reviewer 3 Report

The paper of Loi et al. investigates the 3D bioprinting of a hydrogel based on functionalized gelatin and chitosan as a template for cell's growing. 

The paper is clear, well written and the conclusions are supported by the results. The authors have very well highlighted the originality of their study in the introduction. The materials and the methods are well described and their previous study is cited. In the results section, the authors discussed their obtained results comparing with the literature data. Maybe a FTIR analysis would have been necessary in order to assess the composition of the hydrogel after the fragmentation induced by the cells. Nevertheless, the paper can be published as it is.

Round 2
